# Stearoyl-CoA Desaturases1 Accelerates Non-Small Cell Lung Cancer Metastasis by Promoting Aromatase Expression to Improve Estrogen Synthesis

**DOI:** 10.3390/ijms24076826

**Published:** 2023-04-06

**Authors:** Jiaping Chen, Yangwei Wang, Wangyang Meng, Rong Zhao, Wei Lin, Han Xiao, Yongde Liao

**Affiliations:** Department of Thoracic Surgery, Union Hospital, Tongji Medical College, Huazhong University of Science and Technology, Wuhan 430022, China

**Keywords:** stearoyl-CoA desaturases1, aromatase, non-small cell lung cancer, metastasis, grape seed extract, estrogen, lipid metabolism reprogramming

## Abstract

Metastases contribute to the low survival rate of non-small cell lung cancer (NSCLC) patients. Targeting lipid metabolism for anticancer therapies is attractive. Accumulative evidence shows that stearoyl-CoA desaturases1 (SCD1), a key enzyme in lipid metabolism, enables tumor metastasis and the underlying mechanism remains unknown. In this study, immunohistochemical staining of 96 clinical specimens showed that the expression of SCD1 was increased in tumor tissues (*p* < 0.001). SCD1 knockdown reduced the migration and invasion of HCC827 and PC9 cells in transwell and wound healing assays. Aromatase (CYP19A1) knockdown eliminated cell migration and invasion caused by SCD1 overexpression. Western blotting assays demonstrated that CYP19A1, along with β-catenin protein levels, was reduced in SCD1 knocked-down cells, and estrogen concentration was reduced (*p* < 0.05) in cell culture medium measured by enzyme-linked immunosorbent assay. SCD1 overexpression preserving β-catenin protein stability was evaluated by coimmunoprecipitation and Western blotting. The SCD1 inhibitor A939572, and a potential SCD1 inhibitor, grape seed extract (GSE), significantly inhibited cell migration and invasion by blocking SCD1 and its downstream β-catenin, CYP19A1 expression, and estrogen concentration. In vivo tumor formation assay and a tail vein metastasis model indicated that knockdown of SCD1 blocked tumor growth and metastasis. In conclusion, SCD1 could accelerate metastasis by maintaining the protein stability of β-catenin and then promoting CYP19A1 transcription to improve estrogen synthesis. SCD1 is expected to be a promised therapeutic target, and its novel inhibitor, GSE, has great therapeutic potential in NSCLC.

## 1. Introduction

Lung cancer has high incidence and mortality rates and is a leading cause of cancer-related death worldwide. Non-small cell lung cancer (NSCLC) accounts for approximately 85% of lung cancer cases [1]. The use of low-dose spiral CT is associated with an increased survival rate of lung cancer patients [2]. However, a large proportion of patients (57%) are diagnosed after metastasis and have a low five-year survival rate (6%) [1,3]. The lack of effective drugs to inhibit tumor metastasis has attracted significant attention, and there is an urgent need to develop effective therapeutic approaches for NSCLC patients. An increasing number of tumor metastasis mechanisms have been identified in recent years, including the reprogramming of lipid metabolism [4,5]. Lipid metabolic reprogramming is a newly considered marker of malignancy that promotes rapid proliferation, survival, migration, invasion, and metastasis of tumor cells [6,7]. The dependence of tumor cells on lipid metabolism suggests that inhibiting the expression of lipid metabolism-related proteins may be a potential antitumor metastasis strategy.

Stearoyl-CoA desaturase-1 (SCD1) is the rate-limiting enzyme of lipid synthesis, and is located in the endoplasmic reticulum. SCD1 catalyzes the transformation of saturated fatty acids (SFAs) to monounsaturated fatty acids (MUFAs), which are used to synthesize neutral lipids that are stored in ordered intracellular structures called lipid droplets (LDs) [8,9,10]. SCD1 enzyme activity produces active, lipid-modified Wnt proteins, which are responsible for the activation form of frizzled receptor ligands for the activity of the Wnt/β-catenin signaling [11,12]. Ran et al. reported that SCD1 promotes colorectal cancer metastasis in response to glucose by suppressing PTEN [13]. The feedback activation between nuclear factor κB (NF-κB) and SCD1 is a metabolic marker of ovarian cancer stem cells [14]. SCD1 is downstream of miR-4310, and promotes the growth and metastasis of hepatocellular carcinoma [15]. A metastasis map of human cancer cell lines revealed that SCD1 is the key gene that drives breast cancer metastases to the brain [16]. Previous studies have shown that SCD1 is highly expressed in lung cancer and is associated with a poor prognosis [17,18]. In addition, SCD1 regulates lung cancer stemness via stabilization and nuclear localization of YAP/TAZ [19], promotes the selective apoptosis of ALDH-positive cells [17], and increases cisplatin resistance [20]. However, the mechanism underlying the promotion of lung cancer metastasis by SCD1 still needs to be investigated.

Several recent studies have demonstrated that estrogen is a driver of NSCLC [21]. Estrogens regulate diverse cellular functions, including cell growth, development, and differentiation, by binding to estrogen receptors α, β (ERα and ERβ, respectively), and G-protein coupled ER (GPER1) [22]. As a result, estrogen contributes to lung cancer progression and radio-resistance, particularly in metastasis [23,24]. Adipose tissue is a major source of estrogen, which is a steroid hormone. Many factors in adipose tissue, including inflammatory mediators, adipokines, and cytokines, increase the expression of aromatase (CYP19A1), key enzymes for estrogen synthesis, and local production of estrogen, which contributes to breast cancer development and progression [25]. Several cytokines, such as interleukin-6 (IL-6), oncostatin M, and tumor necrosis factor-α (TNFα), induce CYP19A1 gene expression in lung cancer through the regulation of estrogen synthesis [26]. These studies show a relationship of CYP19A1 expression in adipose tissue with tumor and estrogen synthesis. However, it is unclear whether excessive LD accumulation in lung cancer cells affects aromatase expression and local estrogen synthesis. We have previously reported that estrogen promotes lung cancer progression and metastasis [24,27], and that increased SCD1 expression in tumor tissues is accompanied by increased LDs [18]. Therefore, we hypothesized that the interaction between SCD1, CYP19A1 expression, and estrogen synthesis may play an important role in NSCLC metastasis.

## 2. Results

In the present study, it was hypothesized that SCD1 accelerates non-small cell lung cancer metastasis by promoting aromatase expression to improve estrogen synthesis. The study aimed to prove that SCD1 is an effective therapeutic target to inhibit tumor metastasis, and we preliminarily explored the underlying mechanisms.

### 2.1. SCD1 Is Upregulated in Lung Cancer and Associated with a Poor Prognosis

Our previous study showed that SCD1 was overexpressed in cancer tissues compared to pericancerous tissues, and caused excessive LD accumulation. Here, we report SCD1 in 96 cases of NSCLC tissues and their corresponding pericancerous tissues and tumor metastatic lymph nodes. IHC staining showed a significant difference in the SCD1 expression of tumor tissues and tumor metastatic lymph nodes compared to the pericancerous tissues (*p* < 0.001) (Figure 1A,B). Western blotting analysis was performed to determine the SCD1 expression in four paired pericancerous and matched tumor tissues (Figure 1C,D). Furthermore, the online database KMplotter (http://www.kmplot.com/lung, accessed on 12 September 2021). [28] was used to evaluate the prognostic value of SCD1 mRNA expression (probe 200831sat) in 1925 lung cancer patients. The results demonstrated that a higher SCD1 expression level was associated with a poor prognosis in overall survival (*p* < 0.001, log-rank test) and progression-free survival (*p* < 0.001, log-rank test) (Figure 1E,F).

### 2.2. SCD1 Promotes Migration and Invasion of NSCLC Cells

The significant differential SCD1 expression in NSCLC suggests that SCD1 may be involved in NSCLC invasion and metastasis. To clarify the biological function of SCD1 in NSCLC, we used siRNA transfection and plasmid technology to regulate SCD1 expression in PC9 and HCC827 cell lines (Figure 2A–D). The results showed that SCD1 knockdown could significantly reduce the invasion and migration ability of cell lines; similar results were observed in the transwell (Figure 2E,F) and wound healing (Figure 2I) assays. Conversely, the migration and invasion ability of cell lines was significantly improved following SCD1 overexpression (Figure 2G,H,J). These results suggest that SCD1 is an oncogene involved in the invasion and metastasis of lung cancer cells.

### 2.3. SCD1 Promotes Tumor Invasion and Metastasis via CYP19A1 Upregulation and Enhanced Estrogen Biosynthesis

Because fat is an important biological source of estrogen under physiological conditions, we explored whether SCD1 can promote estrogen biosynthesis in tumor cells. Therefore, we detected the expression level of CYP19A1 in the cases of SCD1 knockdown and overexpression, considering that SCD1 is integral for catalyzing the conversion of androgen to estrogen. The results showed that CYP19A1 had led to constant changes in mRNA and protein levels similar to those with SCD1 expression (Figure 2A–D). These results were observed both in PC9 and HCC827 cell lines. Similarly, we detected the estrogen level in the corresponding cell supernatant culture medium using an ELISA kit. SCD1 knockdown significantly decreased the concentration of extracellular estrogen, while SCD1 overexpression increased the estrogen concentration (Figure 3H,I).

IHC was used to detect the CYP19A1 expression level in tissue samples. The 96 paired tissue samples showed that the CYP19A1 expression level was significantly higher in tumor tissues and metastatic lymph nodes compared to the pericancerous tissues (*p* < 0.001) (Figure 3E,F). We also evaluated the correlation between SCD1 and CYP19A1 (r = 0.3642, *p* < 0.0001, Spearman test) (Figure 3G). Western blot analysis of the four paired tissues also showed similar results (Appendix A).

Furthermore, we explored whether SCD1 promotes the invasion and migration of lung cancer cell lines by upregulating CYP19A1 and estrogen biosynthesis. We constructed an effective siRNA of CYP19A1 (Appendix A). The results of the cotransfection system showed that CYP19A1 knockdown could block the migration and invasion of cell lines caused by SCD1 overexpression (Figure 3L,M and Appendix A). Similar results were obtained from the transwell and wound healing assays. In addition, CYP19A1 knockdown reduced the increase in extracellular estrogen levels caused by SCD1 (Figure 3J,K).

### 2.4. SCD1 Promotes CYP19A1 Expression through the Wnt/β-Catenin Signaling Pathway

The canonical Wnt/β-catenin pathway is significantly activated during tumor invasion and metastasis [29]. The increase in nuclear translocation of β-catenin is the key link to downstream pathway activation. Recent studies have shown that SCD1 promotes β-catenin nuclear translocation of β-catenin and enhances the stemness of lung cancer stem cells [19]. We explored whether SCD1 promotes CYP19A1 expression through the Wnt/β-catenin signaling. For this purpose, the expression level of c-Myc, a critical target gene of β-catenin, was detected when SCD1 was knocked down and overexpressed. A similar relationship was observed between SCD1 expression level and β-catenin with both SCD1 overexpression and knockdown. SCD1 knockdown downregulates the expression of β-catenin and c-Myc, while SCD1 upregulation promotes the expression of β-catenin and c-Myc (Figure 4A,C). IF was used to determine the intracellular localization of β-catenin to regulate SCD1 expression. The results showed that SCD1 promotes β-catenin nuclear translocation (Figure 4B), which is in agreement with our previous results.

Moreover, the treatment of PC9 and HCC827 cells with LiCl, a classic β-catenin signaling activator, induced an increase in CYP19A1 expression (Figure 4D). In contrast, XAV939 treatment led to a decrease in CYP19A1 mRNA and protein levels (Figure 4E and Appendix A). It also blocked the increase in CYP19A1 expression induced by SCD1 overexpression (Figure 4E).

We wondered whether CYP19A1 or estrogen could affect SCD1 expression and the Wnt/β-catenin signaling. The results show that CYP19A1 knockdown leads to a decrease in SCD1 mRNA and protein levels (Appendix A), while CYP19A1 overexpression promotes an increase in SCD1 mRNA and protein levels in PC9 and HCC827 cell lines (Appendix A). The protein levels of β-catenin, CYP19A1, and SCD1 were significantly increased when exposed to estrogen at a concentration of 10 nM for 12, 24, and 48 h in PC9 and HCC827 cell lines (Appendix A). Similar results were observed after treating PC9 and HCC827 cells with 0, 10, and 50, 100 nM estrogen for 48 h (Appendix A). Based on current findings, we believe that there is a feedback activation mechanism between estrogen and SCD1.

IHC showed that β-catenin expression in tissue samples and the staining of tumor tissues and metastatic lymph nodes were significantly stronger than those in the adjacent normal tissues (*p* < 0.001) (Figure 4F,G). In addition, we mapped the relative expression levels of SCD1 and β-catenin in tumor tissues. The results showed a significant linear positive correlation between the expression levels of SCD1 and β-catenin (r = 0.2740, *p* < 0.0001, Spearman test) (Figure 4H); the expression of β-catenin and CYP19A1 showed similar results (r = 0.2470, *p* < 0.0001, Spearman test) (Figure 4I).

Lipid modification of Wnts is necessary for Wnt protein secretion and subsequent signal pathway activation [30]. It causes a decrease in β-catenin phosphorylation and the subsequent ubiquitination-mediated degradation of the lysosomal pathway. In contrast, it increases β-catenin accumulation and nuclear translocation. We explored whether SCD1 leads to a decrease in β-catenin ubiquitination level and causes catenin accumulation. A vector and SCD1 plasmid were transfected into pC9 cells and treated with DMSO. CHX was used to detect the β-catenin protein stability at different time points (0, 0.5, 1, 2, 4, and 6 h after drug intervention). As shown in Figure 4J, the β-catenin protein stability in PC9 cells transfected with the vector was significantly reduced (as indicated by more rapid depletion) compared to PC9 cells transfected with SCD1. Furthermore, the β-catenin protein expression of PC9 cells transfected with the vector decreased within four hours of treatment with MG132, while no significant decrease was observed in cells transfected with SCD1 (Figure 4J). Subsequently, we evaluated whether the ubiquitination of β-catenin was involved in proteasome-mediated β-catenin degradation. PC9 cells were transfected with a vector or SCD1 plasmid and incubated with MG132 for 12 h. Then, the total protein lysates were immunoprecipitated with β-catenin antibody and immunoblotted with antiubiquitin antibody. As shown in Figure 4K, compared to PC9 cells transfected with the vector, the immunoprecipitated β-catenin PC9 cells transfected with SCD1 clearly showed that the polyubiquitination pattern of β-catenin was significantly reduced.

The above results suggest that SCD1 promotes CYP19A1 expression by activating the β-catenin signaling pathway, and SCD1 is involved in the stabilization of β-catenin.

### 2.5. SCD1 Inhibitor A939572 Effectively Inhibits the β-Catenin/CYP19A1 Axis and Estrogen Biogenesis

We explored whether A939572 can effectively block the SCD/β-catenin/CYP19A1 axis and estrogen biosynthesis. At concentrations of 1 and 10 μM, A939572 was added to the supernatant of pC9 and 827 cell line culture medium. SCD/β-catenin/CYP19A1 axis was significantly inhibited at a concentration of 1 μM, and the effect was markedly enhanced at 10 μM (Figure 5A,B). The estrogen concentration in the supernatant of the cell culture medium was also decreased after A939572 treatment (Figure 5C,D). The invasion and migration of pC9 and HCC827 cell lines exposed to A939572 at concentrations of 1 and 10 μM were significantly inhibited (Figure 5E–H,J,K and Appendix A). Furthermore, oil red O staining and Nile red staining showed a significant reduction in intracellular lipids 48 h after A939572 treatment at 1 and 10 μM (Figure 5I–K).

### 2.6. Grape Seed Extract Is a Potential Inhibitor of SCD1 and Inhibits Lipid Accumulation and Estrogen Biosynthesis

Grape seed extract (GSE) is rich in flavonoids and oligoprocyanidins (OPCs), and shows antitumor activity in vitro [31]. GSE treatment significantly inhibited the proliferation and migration of tumor cells in a dose-dependent manner. It also showed considerable antioxidant activities, regulated lipid homeostasis [32], and inhibited aromatase gene expression [33]. We investigated whether GSE attenuates the migration and invasion abilities of PC9 and HCC827 cell lines by inhibiting the SCD1/β-catenin/CYP19A1 axis. We examined the effect of GSE on SCD1 expression, and Western blot experiments showed that treatment with 10–100 μg/mL of GSE suppressed the expression levels of SCD1 and the downstream β-catenin, CYP19A1, and c-Myc (Figure 6A,B). We detected the estrogen concentration in the cell culture media treated with 20 and 50 μg/mL of GSE, and found that GSE could effectively inhibit estrogen synthesis (Figure 6C,D). Furthermore, oil red O staining and Nile red staining showed a significant reduction in intracellular lipids 48 h after GSE treatment at 20 and 50 μg/mL (Figure 6K–N). GSE treatment also markedly reduced the migration and invasion ability of PC9 and HCC827 cell lines (Figure 6E–H and Appendix A).

### 2.7. In Vivo shSCD1 Inhibits Tumor Growth and Metastasis

We explored whether SCD1 could promote tumor growth and metastasis in animal models. We used expression lentivirus and shRNA to construct the cell lines with stable knockdown of SCD1 in PC9 cells (Appendix A), and constructed a subcutaneous transplanted tumor model in nude mice. The results showed that SCD1 knockdown significantly reduced the growth of tumor volume and weight over 18 days (Figure 7A–C). A939572 treatment also leads to a decrease in tumor growth (Appendix A). Based on the effect of SCD1 on LD accumulation in the above study, oil red O staining was used, which showed that the LD content in tissues was significantly decreased after SCD1 knockdown in subcutaneous tumors (Figure 7D). IHC also showed that SCD1, β-catenin, CYP19A1, and KI67, markers of malignancy, were significantly decreased in the SCD1 knockdown group (Figure 7E). In vivo imaging showed that SCD1 knockdown significantly reduced the metastasis of tumor cells in the tail vein metastasis model over 49 days (Figure 7G), especially liver and lung metastases. Subsequently, H&E staining was used as a visual indicator of liver and lung metastases (Figure 7F).

## 3. Discussion

Lipid metabolism reprogramming, which increases lipid uptake, storage, and production, occurs in various cancers and contributes to rapid tumor growth [34,35,36]. SCD1 is a key oncogene involved in fatty acid synthesis, which leads to abnormal lipid accumulation. The specific mechanism role of SCD1 in regulating NSCLC metastasis is not fully understood. Our experimental demonstrate that SCD1 promotes NSCLC metastasis by activating the Wnt/β-catenin pathway to improve CYP19A1 expression and estrogen synthesis. Moreover, our results indicate that GSE is a potentially effective SCD1 inhibitor that prevents lipid accumulation and blocks tumor cell invasion and migration via decreasing CYP19A1 expression and estrogen synthesis.

Abnormal activation of the Wnt/β-catenin signaling pathway contributes to the progression of various cancers, such as osteosarcoma, colorectal, breast, lung, and hepatocellular cancers [30]. The canonical Wnt pathway, also known as Wnt/β-catenin signaling pathway, involves the nuclear translocation of β-catenin and activation of target genes via TCF/LEF transcription factors. Several factors have been reported to regulate the activation of Wnt/β-catenin signaling, such as syndecan-1 and ADNP [37,38]. Ras/Raf/Mek/Erk signaling and TGF-β signaling contribute to the activation of Wnt/β-catenin signaling [39]. Evidence shows that SCD1 is an activator of β-catenin, which induces β-catenin nuclear localization. This may be caused by MUFA, the direct product of SCD1, since lipid modification on Wnt proteins is required for its secretion. Our study indicated that SCD1 promotes β-catenin accumulation by improving protein stability, reducing the lysosome and proteasome-mediated β-catenin degradation.

Estrogen receptor-mediated signaling pathways are important in tumor pathogenesis. The crosstalk between the estrogen signaling pathway and Wnt/β-catenin is controversial [40,41,42]. The activation of ER-α inhibits the Wnt/β-catenin pathway and the transcription of its downstream target genes, c-Myc and Cyclin D1, in hepatocarcinogenesis [42]. Estrogen induces Wnt-activated self-renewal of breast cancer stem cells via Erα [41]. The origin of local estrogen in breast cancer is of wide concern; however, the mechanism of local estrogen production in lung cancer is not clear. β-catenin is an essential transcriptional regulator of CYP19A1 and is involved in regulating the CYP19A1 expression and the production of estrogen in granulosa cells [43]. In this study, we demonstrated that the SCD1-activated Wnt pathway is an important regulator of CYP19A1 and an important source of local estrogen in NSCLC. Our study combines fatty acid metabolism with the estrogen signaling pathway.

Lipid accumulation and overexpression of lipid synthesis genes in tumor cells may lead to worsening of the inflammatory environment and reactive oxygen species [44,45]. GSE is a commonly used dietary antioxidant that shows anticancer activity in a variety of tumors and may reduce obesity [46,47]. Similarly, GSE treatment led to the downregulation of two transcription factors: cyclic AMP-responsive element-binding protein-1 (CREB-1) and glucocorticoid receptor (GR), which upregulate the aromatase gene expression through the promoters I.3/II and I.4 in breast cancer [33]. Our present experiments indicated that GSE significantly blocking NSCLC metastasis is partly achieved by inhibiting SCD1-mediated CYP19A1 expression and estrogen synthesis. GSE is recognized as a novel inhibitor of SCD1.

There is increasing evidence that estrogen is directly related to NSCLC. CYP19A1 is necessary for estrogen synthesis in tumors and is highly expressed in lung tumors [48,49]. Preclinical data have confirmed that estrogen induces lung cancer. There is a lack of sufficient research on local CYP19A1 expression and estrogen production in lung cancer. This study combines the lipid metabolism pathway in tumor cells with estrogen production to clarify the pathway of local estrogen production in lung cancer. Moreover, estrogen also leads to an increase in SCD1 expression; there is a feedback activation mechanism between estrogen and SCD1. However, which form of ER is involved in estrogen’s regulation of SCD1 expression and the specific mechanisms remain an enigma. In fact, SCD1 expression is affected by multiple factors, such as epidermal growth factor receptor (EGFR), STK11/KEAP1 comutation, NF-κB, and sterol regulatory element binding protein 1 (SREBP1) [14,50,51,52]. Therefore, we can continue to conduct in-depth research in related directions.

This study demonstrates that SCD1 is a promising target for inhibiting NSCLC metastasis. However, the effects of SCD1 on NSCLC are broad. For example, SCD1 expression in tumor cells and immune cells can cause immune tolerance, and its inhibition can enhance the therapeutic efficacy of antitumor T cells and anti-PD-1 antibodies [53]. SCD1 inhibition induces ferroptosis in STK11/KEAP1-comutated lung cancer cells and blocks tumor growth [50]. In lung cancer cells with EGFR mutations, the combinatorial treatment modality of SCD1 and EGFR-tyrosine kinase inhibitors (TKIs) showed better antitumor effects [18,54]. These studies demonstrate that the promotion of lung cancer by SCD1 is multifaceted; SCD1 inhibition may block lung cancer progression through multiple pathways. Our study elucidates a novel mechanism by which SCD1 promotes NSCLC progression and may provide a part of the evidence to drive the clinical research of SCD1 inhibitors.

## 4. Materials and Methods

### 4.1. Patients and Samples

A total of 96 formalin-fixed, paraffin-embedded tissue samples were collected from primary NSCLC patients diagnosed at the Department of Thoracic Surgery of Union Hospital affiliated with Tongji Medical College of Huazhong University of Science and Technology (Wuhan, China) between 2014 and 2018. The patients did not have a history of use of chemotherapy, radiotherapy, or hormone therapy preoperatively. The clinical information of patients was recorded, and the diagnosis was confirmed by at least two pathologists. The pTNM stage and tumor differentiation grade were recorded at the Union Hospital. The tissue samples were acquired after obtaining informed consent from patients. Table 1 presents the baseline characteristics of study patients.

### 4.2. Cell Lines and Cell Culture

The NSCLC cell lines PC-9 and HCC827 were originally obtained from ATCC and the Chinese Academy of Sciences (Shanghai, China) prior to 2014. All cell lines were cultured in RPMI-1640 medium (Servicebio, Wuhan, China) supplemented with 10% fetal bovine serum (FBS; Gibco, Bethesda, MD, USA) and antibiotics (100 units/mL penicillin and 100 μg/mL streptomycin). All cell lines were cultured at 37 °C in a humidified atmosphere with 5% CO_2_ and 95% air.

### 4.3. Cell Treatment and Cell Morphological Observation

LiCl, an activator of β-catenin, was purchased from Co. (L9650, Sigma-Aldrich, St. Louis, MO, USA). Cell culture experiments were performed using reagents formulated in 100% dimethyl sulfoxide (DMSO). XAV-939 was purchased from MedChemExpress (MCE) (HY-15147, USA). A939572 (HY-50709, USA), GSE (HY-N7072, USA), MG-132 (HY-13259, USA), estrogen (Estradiol, HY-B0141, USA), and Cycloheximide (CHX, HY-12320, USA) were purchased from MCE.

Identical amounts of cells were planted in a 6-well plate and treated for 48 h. The cells were cultured at 37 °C in a humidified atmosphere containing 5% carbon dioxide and 95% air. After 48 h of treatment, the cell morphology was observed by an inverted phase-contrast microscope (Olympus, Tokyo, Japan).

### 4.4. Oil Red O Staining

We prepared an oil red O working solution (60% oil red O stock solution (BA-4081; Baso, Zhuhai, China) and 40% deionized water). The cells were cultured in a six-well plate to reach a confluency of 30%. The cell culture medium was removed, and the cells were washed twice with PBS. The cells were fixed in 4% paraformaldehyde. Then, paraformaldehyde was removed, and the cells were allowed to dry naturally for 10 min. The cells were mixed with the oil red O working solution and stained at room temperature (RT) for 30 min. Then, phosphate-buffered saline (PBS) was used to remove the excess oil red stain. The cells were dried and observed under a microscope. Frozen cancer tissues were embedded in the OCT compound (Sakura, Tokyo, Japan) and cut into 10 μm sections. The slides were washed several times with distilled water and precultured in 60% isopropanol. Finally, the slides were stained with filtered oil red O working solution. After a series of washing steps in 60% isopropanol, the nuclei were counterstained with hematoxylin and eosin (H&E) prior to differentiation in 1% hydrochloric acid and alcohol. After washing several times with distilled water, the slides were sealed with glycerin/gelatin.

### 4.5. Nile Red Staining

Living cells were inoculated in a six-well plate, fixed in 4% paraformaldehyde for 20 min at RT, and incubated with Nile red (HY-D0718, MCE, USA) in PBS at a ratio of 1:2000 for 30 min at 37 °C. After washing three times with PBS, the cells were counterstained with DAPI (GDP1024; Servicebio, Wuhan, China) for 5 min at RT. The cells were observed under a fluorescence microscope (Olympus, Tokyo, Japan). The representative images were obtained from three independent experiments.

### 4.6. IHC

Formalin-fixed, paraffin-embedded tissue was cut into 4 μm sections and analyzed by IHC, as described previously [27]. In brief, anti-SCD1 (diluted 1:100, Cell Signaling Technology, Danvers, MA, USA, catalog number: #2794), anti-β-catenin (diluted 1:100, Abclonal, Woburn, MA, USA, catalog number: A19657), anti-CYP19A1 (diluted 1:100, Abclonal, catalog number: A2161), and anti-Ki67 (diluted 1:500, Servicebio, catalog number: GB111141) were used as primary antibodies to incubate the tissue sections after heat-induced epitope retrieval (in 10 mM sodium citrate buffer of pH 9.0), then incubated with secondary antibody (diluted 1:100, G1213, Servicebio), followed by diaminobenzidine. The slides were counterstained with hematoxylin. The researchers scored the slides for the expression level of SCD1, β-catenin, and CYP19A1 in a double-blind manner, and the scoring system was reasonably adjusted according to the previous description [27].

### 4.7. Coimmunoprecipitation (Co-IP) and Western Blotting

Ground tumor tissues and harvested cells were fully lysed with ice-cold RIPA lysis buffer (1% Nonidet P-40, 50 mM Tris-HCl (pH 7.4), 150 mM sodium chloride, and 0.5% sodium deoxycholate) which was mixed with 1% protease inhibitor and phosphatase inhibitor. Followed by centrifuging for 20 min at 12,000 rpm/min at 4 °C. The protein in the supernatant was quantified by Bradford [55]. For the Co-IP assay, cells were pretreated with MG132 for 10 h and lysed using 1% Nonidet P-40. Equivalents of 1 mg protein in the soluble fraction were adjusted for Co-IP in 0.5 mL of total volume following incubation with 5 uL anti-β-catenin (Abclonal, catalog number: A19657) antibody. Normal IgG (IgG) was used as an IP control. IP was performed with 50 uL of a slurry with Protein A/G Magnetic Beads (HY-K0202, MCE) for another 2 h. Beads were then removed and washed with lysis buffer. The protein solution was separated by 10%SDS-PAGE gel. The gel was transferred to a nitrocellulose membrane. Following incubation with 5% bovine serum albumin (BSA) for 60 min, the membranes were cut into strips according to the different molecular weights. The membrane was incubated with the corresponding primary antibodies against SCD1 (diluted 1:1000, Cell Signaling Technology, catalog number: #2794), β-catenin (diluted 1:1000, Abclonal, catalog number: A19657), CYP19A1 (diluted 1:1000, Abclonal, catalog number: A2161), c-Myc ((diluted 1:1000, Abmart, Berkeley Heights, NJ, USA, catalog number: 10494–1-AP TA0358), GAPDH (diluted 1:1000, Proteintech, catalog number: 10494–1-AP), β-actin (diluted 1:1000, Cell Signaling Technology, catalog number: #3700) Ubiquitin (diluted 1:1000, Cell Signaling Technology, catalog number: #58395) overnight at 4 °C, then subsequently incubated with HRP-conjugated secondary antibodies for 1 h at RT. ECL detection system (Bio-Rad Laboratories, Hercules, CA, USA) was used for imaging.

### 4.8. Immunofluorescence (IF) Staining of Cultured Cells

Cells cultured in cover slides were transfected for 48 h. The cells were fixed in 4% paraformaldehyde for 15 min, incubated with 0.5% triton-100× for 15 min, and incubated with 5% BSA for 30 min at RT and with anti-β-catenin (diluted 1:100, Abclonal, catalog number: A19657) overnight at 4 °C. Alexa Fluor^®^ 488-conjugated Goat IgG (H + L) was used to label the cells at RT for 1 h in a dark area, followed by incubation with DAPI for 5 min. The cells were visualized under a fluorescence microscope (Olympus).

### 4.9. RNA Extraction and qRT-PCR

The cells’ total RNA was extracted using Trelief^TM^ RNAprep FastPure Tissue & Cell Kit (TSP413, Tsingke, Beijing, China). NanoDrop 2000 spectrophotometer (NanoDrop Technologies, Wilmington, DE, USA) was used for RNA purity and concentration detection. 1000 ng of the product was used for the subsequent reverse transcription process, which was then performed PCR by StepOnePlus Real-Time PCR System (Thermo Fisher Scientific, Waltham, MA, USA) with SYBR Green mix (Servicebio, Wuhan, China). With endogenous GAPDH as the control, all qPCR processes were repeated three times. The primers were obtained from TSINGKE, the primer sequences for qPCR were as follows: SCD1: Forward: 5′-TCT AGC TCC TAT ACC ACC ACC A-3′, Reverse: 3′-TCG TCT CCA ACT TAT CTC CTC C-5′, CYP19A1: Forward: 5′-TGG AAA TGC TGA ACC CGA TAC-3′, Reverse: 3′-AAT TCC CAT GCA GTA GCC AGG-3′, GAPDH: forward: 5′-CGT GGA AGG ACT CAT GAC CA-3′, reverse: 5′-GCC ATC ACG CCA CAG TTT C-3′.

### 4.10. Wound Healing Assays

Equal amounts of pretreated cells were planted on each side of Culture-Insert 2 Well (No:80209, Ibidi, Gräfelfing, Germany) until fusion reached 100%. The Culture-Insert was removed and cells were further cultured in serum-free medium. The wounds were observed at 24 h using an inverted microscope (Olympus, Tokyo, Japan).

### 4.11. Transwell Assays

The cells were treated with drugs or transfected for 48 h, and the above cells were added to the top chamber of the transwell chamber. On this basis, Matrigel with a ratio of 1:8 was added for the invasion assay. Incubation was performed for 24 h, then fixed with paraformaldehyde for 15 min following incubation with crystal violet dye for 10 min. The resulting image was taken using a microscope. The number of cells used for migration analysis was 5 × 10^4^, while that was double for invasion analysis.

### 4.12. Cell Transfections

Lipofectamine 3000 reagents (Thermo-Fisher Scientific, Waltham, MA, USA) were used for plasmid and siRNA transfections, with 1.5 μg of plasmid DNA per well and 40 nM of final siRNA concentration in the 6-well dish. According to the protocol recommended by the reagent manufacturer, the transfection mixture was fully mixed before adding dropwise to cells and incubated at RT for 15 min. The culture medium was replaced with fresh medium after 6 h incubation at 37 °C. The siRNA sequence used was siSCD1: 5′-CCC ACC UAC AAG GAU AAG GAA TT-3′, siCYP19A1: 5′-GGG UAU AUG GAG AAU UCA U dTdT-3′. The same sequence was used for lentiviral-RNAi contraction provided by Genechem (Shanghai, China), which was used for cell transfection following continuous puromycin treatment for 3 days to select a stable cell line.

### 4.13. Enzyme-Linked Immunosorbent Assay (ELISA)

The estrogen concentration in the culture medium was determined using the ELISA kit according to the manufacturer’s protocol (Cusabio Biotech, Wuhan, China, catalog number: CSB-E07286h). For statistical analysis, the culture medium was independently collected three times.

### 4.14. Tumor Formation Assay

A total of 4 × 10^6^ PC9 cells infected with shSCD1 or negative control shRNA were subcutaneously injected into 6-week-old nude mice (Vital River, Beijing, China). Mice were euthanized 18 days after cell implantation, and the tumor weight was measured. Tumor growth was measured using a digital caliper every 3 days for 18 days.

### 4.15. In Vivo Cancer Metastasis Assay

The metastatic ability of tumor cells was evaluated using a tail vein metastasis model in nude mice. Notably, 2 × 10^6^ PC9 cells infected with the negative control shRNA, shSCD1, were injected into the tail vein of mice. All mice were euthanized. After 7 weeks of observation, the liver and lung tissues were fixed, embedded in paraffin, sectioned, and stained with H&E.

### 4.16. Statistical Analysis

Unpaired two-tailed Student’s *t*-test and one-way analysis of variance were performed for intergroup comparisons. For cell-based assays, differences between groups were assessed using two-tailed Student’s *t*-tests, unless indicated otherwise. Data were analyzed using Graphpad-Prism 9.03 statistics software. Each experiment was repeated at least three times with comparable results unless indicated otherwise. The relationship between SCD1 expression level and the prognosis of lung cancer patients was analyzed using the online database KMplotter (http://www.kmplot.com/lung, accessed on 12 September 2021) [28]. All results are presented as mean  ±  SD (standard deviation). *p*-value < 0.05 was considered statistically significant.

## 5. Conclusions

In conclusion, searching for reliable treatment methods to inhibit tumor metastasis may benefit lung cancer patients. In recent years, advances in the understanding of tumor metabolism have demonstrated the need for complex metabolic rewiring in tumor cells to adapt to adverse survival conditions. Our results reveal a relationship between abnormal lipid metabolism and local estrogen production in lung cancer. That is, our study identified that SCD1 drives lung cancer metastasis by promoting CYP19A1 expression and estrogen production for the first time. SCD1 is expected to be a promising therapeutic target in NSCLC. In the future, the novel inhibitor GSE may have the potential for SCD1 gene targeting therapy.

## Figures and Tables

**Figure 1 ijms-24-06826-f001:**
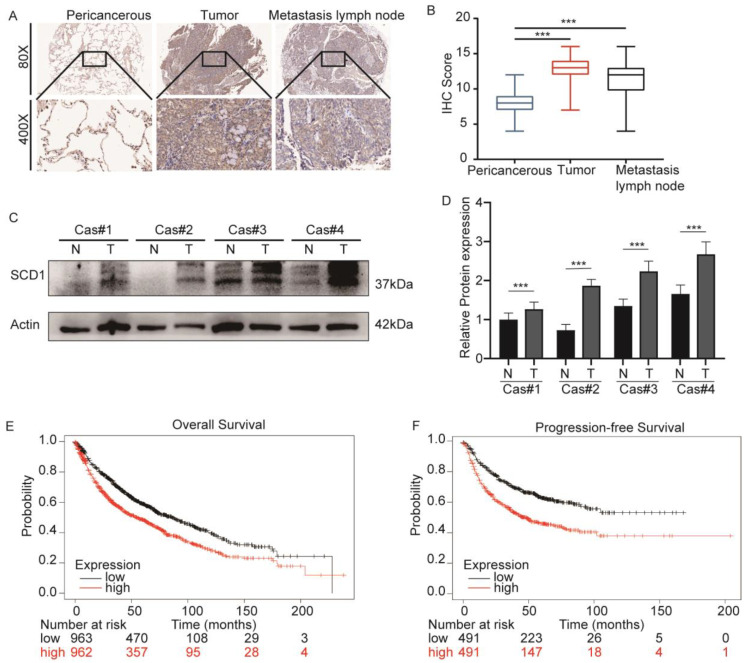
SCD1 is upregulated in lung cancer and is associated with poor prognosis. (**A**) IHC analyses of 96 cases of cancer tissues and their corresponding pericancerous tissues and tumor metastatic lymph nodes. (**B**) IHC scores of 96 cases. (**C**,**D**) The SCD1 protein expression in lung cancer tissues and corresponding pericancerous tissues was evaluated by Western blotting. The prognostic value of SCD1 mRNA expression in lung cancer. (**E**) Overall survival (OS) and (**F**) progression-free survival (PFS). *** *p* < 0.001.

**Figure 2 ijms-24-06826-f002:**
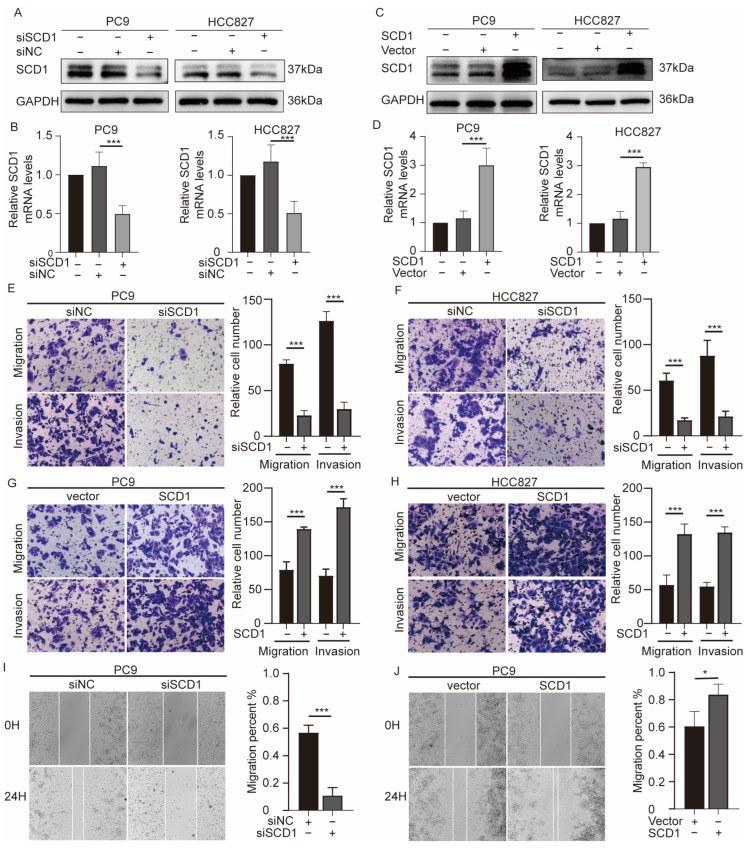
SCD1 promotes NSCLC cell migration and invasion. SCD1 (**A**,**C**) protein, and (**B**,**D**) mRNA (*n* = 3) levels were verified by Western blotting and qRT-PCR in PC9 and HCC827 cells with transient SCD1 knockdown and overexpression. (**E**–**H**) Cell migration and invasion ability of transfected PC9 and HCC827 cells were evaluated using transwell assays (magnification: 200×, *n* = 3 per group). (**I**,**J**) The migration ability of transfected PC9 cells assessed using wound healing assays (magnification: 100×). * *p* < 0.05; *** *p* < 0.001. Error bars indicate mean ± SD.

**Figure 3 ijms-24-06826-f003:**
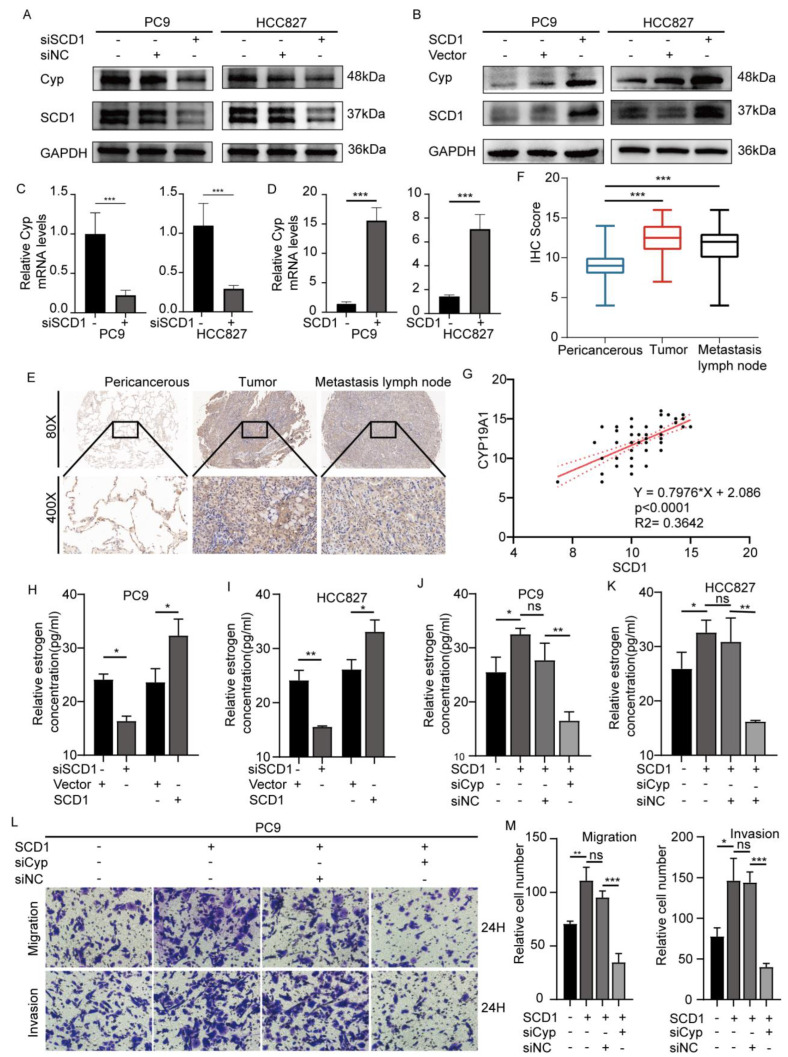
SCD1 promotes tumor invasion and metastasis via upregulating CYP19A1 expression and estrogen biosynthesis. CYP19A1 (**A**,**B**) protein, and (**C**,**D**) mRNA (*n* = 3) levels were evaluated using Western blotting and qRT-PCR in PC9 and HCC827 cells with transient SCD1 knockdown and overexpression. (**E**) IHC analyses of CYP19A1 of 96 cases of cancer tissues and their corresponding pericancerous tissues and tumor metastatic lymph nodes. (**F**) IHC scores of 96 cases. (**G**) The relative CYP19A1 expression level was plotted against the relative SCD1 expression level in 96 samples (r [Spearman] = 0.3642; *p* < 0.0001). The CYP19A1 protein level in PC9 and HCC827 cells with transient CYP19A1 knockdown. Relative estrogen concentration of cell supernatant culture medium with transient SCD1 knockdown and overexpression in (**H**) PC9 and (**I**) HCC827 cell lines. The relative estrogen concentration of cell supernatant culture medium with transient SCD1 overexpression and CYP19A1 knockdown in (**J**) PC9 and (**K**) HCC827 cell lines. (**L**,**M**) The cell migration and invasion ability of transfected PC9 and HCC827 cells were evaluated using transwell assays (magnification: 200×, *n* = 3 per group). * *p* < 0.05; ** *p* < 0.01; *** *p* < 0.001; ns: no significance. Error bars indicate mean ± SD.

**Figure 4 ijms-24-06826-f004:**
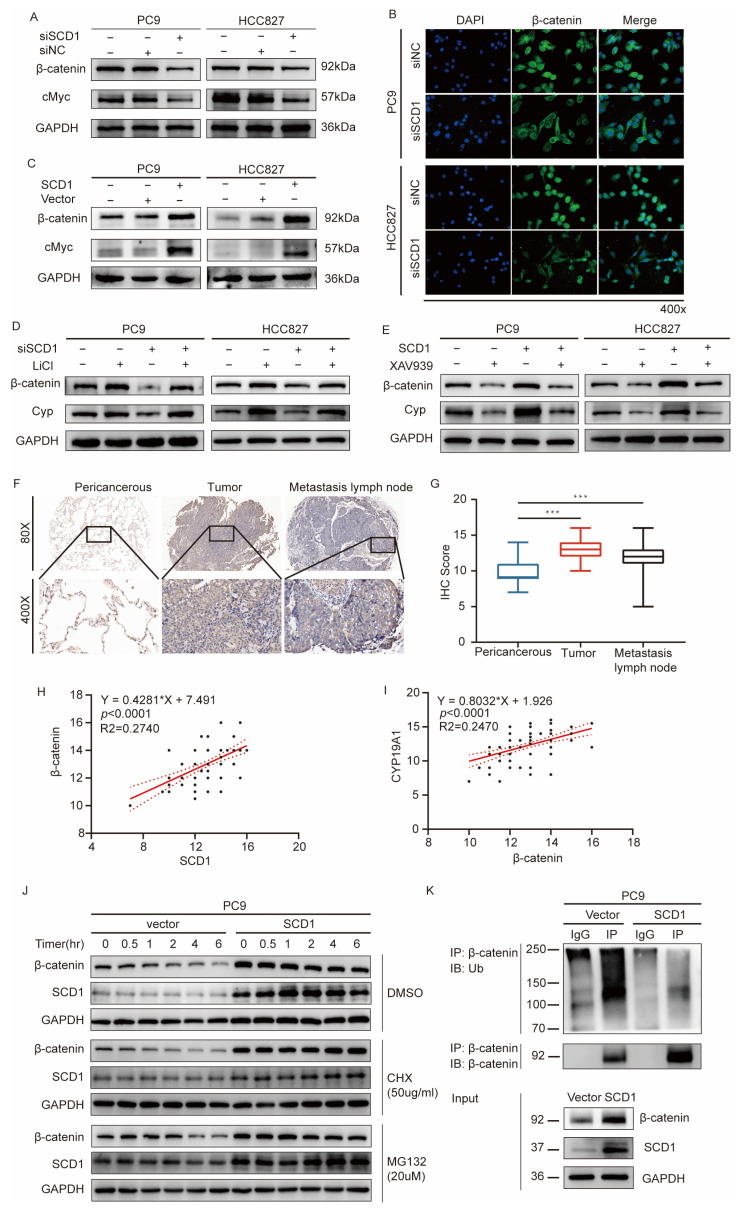
SCD1 promotes CYP19A1 expression through the Wnt/β-catenin signaling pathway. (**A**,**C**) The protein levels of β-catenin and c-Myc were verified using Western blotting in PC9 and HCC827 cells with transient SCD1 knockdown and overexpression. (**B**) The intracellular localization of β-catenin was verified by IF in PC9 and HCC827 cells with transient SCD1 knockdown (magnification: 400×). (**D**) The protein levels of β-catenin and Cyp were verified by Western blotting in PC9 and HCC827 cells with transient SCD1 knockdown and LiCl treatment. (**E**) The protein levels of β-catenin and Cyp were verified by Western blotting in PC9 and HCC827 cells with transient SCD1 overexpression and XAV939 treatment at 15 μM. (**F**) IHC analyses of β-catenin of 96 cases of cancer tissues and their corresponding pericancerous tissues and tumor metastatic lymph nodes. (**G**) IHC scores of 96 cases. (**H**) Relative β-catenin expression level was plotted against the relative SCD1 expression level in 96 samples (r [Spearman] = 0.2740; *p* < 0.0001). (**I**) The relative CYP19A1 expression level was plotted against the relative β-catenin expression level in 96 samples (r [Spearman] = 0.2470; *p* < 0.0001). (**J**) The β-catenin protein levels were verified using Western blotting in PC9 cells with transient SCD1 overexpression and treatment with DMSO, CHX, and MG132 for 0, 0.5, 1.0, 2.0, 4.0, and 6.0 h. (**K**) IB: Ub was verified by Western blotting in PC9 cells with transient SCD1 overexpression and MG132 treatment for 12 h. *** *p* < 0.001. Ub: ubiquitin. Cyp: CYP19A1.

**Figure 5 ijms-24-06826-f005:**
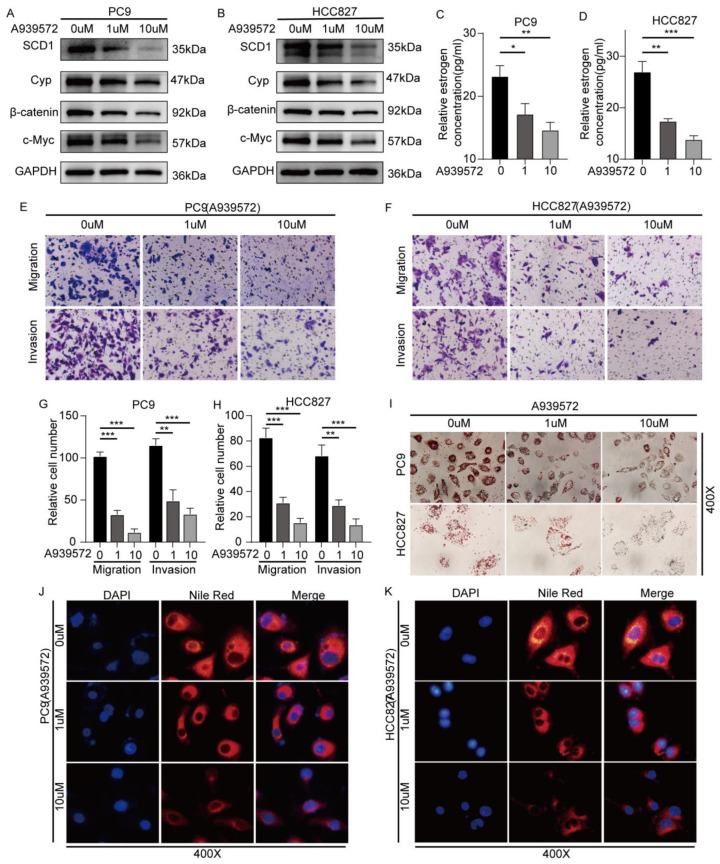
The SCD1 inhibitor A939572 effectively inhibits the β-catenin/CYP19A1 axis and estrogen biogenesis. The protein levels of SCD1, Cyp, β-catenin, and c-Myc after A939572 treatment at concentrations of 0, 1, and 10 μM in (**A**) PC9 and (**B**) HCC827 cells. The relative estrogen concentration of cell supernatant culture medium with A939572 treatment at concentrations of 0, 1, and 10 μM for 48 h in (**C**) PC9 and (**D**) HCC827 cells. The cell migration and invasion ability of A939572 treatment at concentrations of 0, 1, and 10 μM for 48 h in (**E**,**G**) PC9 and (**F**,**H**) HCC827 cells were evaluated using transwell assays (magnification: 200×, *n* = 3 per group). (**I**) Oil red O staining of PC9 and HCC827 cells treated with A939572 for 48 h. Nile red staining of (**J**) PC9 and (**K**) HCC827 cells treated with A939572 for 48 h. * *p* < 0.05; ** *p* < 0.01; *** *p* < 0.001. Error bars indicate mean ± SD. Cyp: CYP19A1.

**Figure 6 ijms-24-06826-f006:**
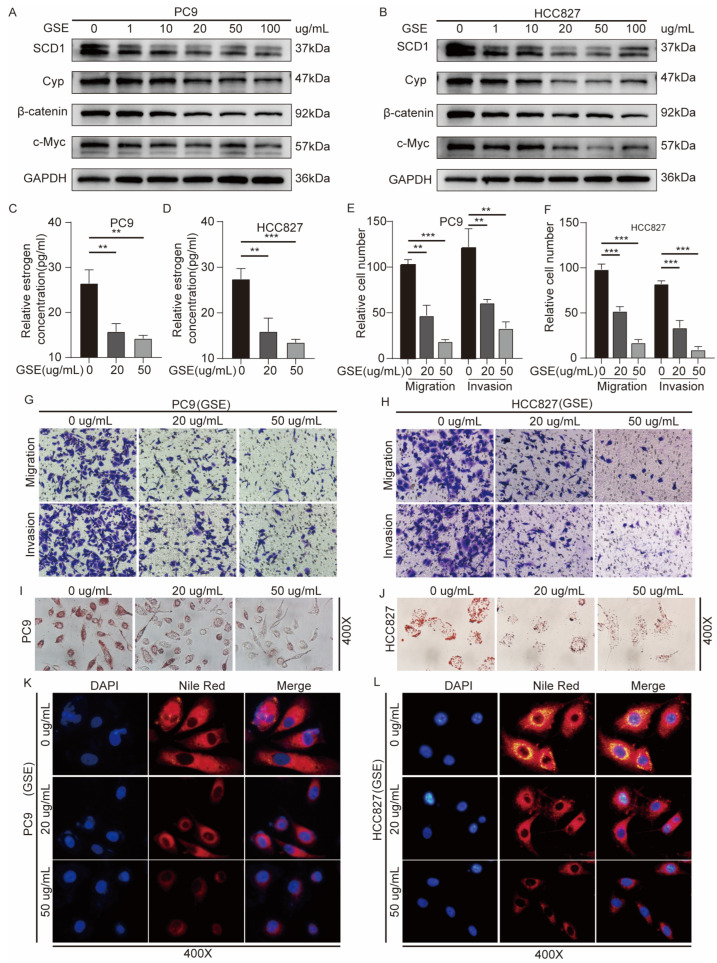
GSE acts as a potential inhibitor of SCD1 and inhibits lipid accumulation and estrogen biosynthesis. The protein levels of SCD1, Cyp, β-catenin, c-Myc with GSE treatment at concentrations of 0, 1, 10, 20, 50, and 100 μg/mL in (**A**) PC9 and (**B**) HCC827 cells. The relative estrogen concentration of cell supernatant culture medium with GSE treatment at concentrations of 0, 20, and 50 μg/mL for 48 h in (**C**) PC9 and (**D**) HCC827 cells. The cell migration and invasion ability of GSE treatment for 48 h in (**E**,**G**) PC9 and (**F**,**H**) HCC827 cells were evaluated using transwell assays (magnification: 200×, *n* = 3 per group). The oil red O staining of (**I**) PC9 and (**J**) HCC827 cells with GSE treatment for 48 h. Nile red staining of (**K**) PC9 and (**L**) HCC827 treatment with GSE for 48 h. ** *p* < 0.01; *** *p* < 0.001. Error bars indicate mean ± SD. Cyp: CYP19A1. GSE: Grape seed extract.

**Figure 7 ijms-24-06826-f007:**
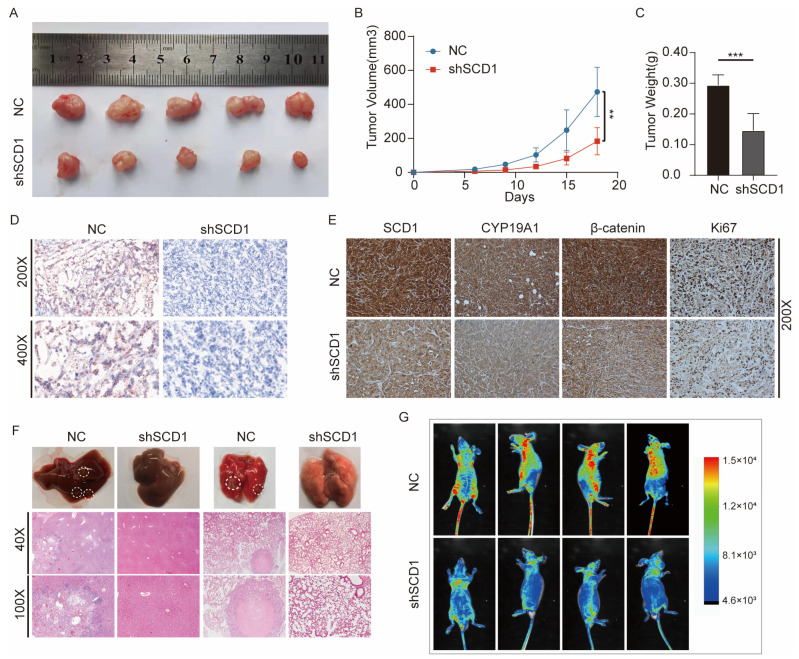
shSCD1 inhibits tumor growth and metastasis in vivo. (**A**–**C**) PC9 cells with stable SCD1 knockdown were injected into nude mice. The tumor size was measured after every 3 days. The data are expressed as mean ± SEM of different tumors in each group. Tumor images were obtained from mice. The relationship between tumor size (mm^3^) and the number of days after tumor cell implantation was evaluated. At the end of the experiment, the tumor was weighed after resection (*n* = 5) (independent samples *t*-test). The tumor size was measured every 3 days, and the final measurement was performed on the 18th day. (**D**) Oil red O staining of the tumor xenografts from the SCD1 knockdown and control groups. (**E**) Immunohistochemical (IHC) staining for SCD1, β-catenin, CYP19A1, and the marker of tumor malignancy (KI67) in tumor xenografts. (**F**) H&E staining of the liver (left) and lung (right) tissues from SCD1 knockdown and control groups. (**G**) Live fluorescence images of SCD1 knockdown in the metastasis model group and control group over 49 days. ** *p* < 0.01; *** *p* < 0.001.

**Table 1 ijms-24-06826-t001:** Relationship between SCD1/CYP19A1 expression levels and clinical characteristics of NSCLC patients.

	No. of Patients (%)	SCD1 Expression		χ2	*p*-Value	CYP19A1 Expression		χ2	*p*-Value
		High	Low			High	Low		
All	96	83	13			71	25		
Genders									
Females	23	17	6	4.066	0.044	17	6	0.000	0.996
Males	73	66	7			54	19		
Age									
<60	58	50	8	0.008	0.929	41	17	0.813	0.367
≥60	38	33	5			30	8		
T stage									
1–2	64	57	7	0.429	0.513	45	19	1.325	0.249
3–4	32	27	5			26	6		
N stage									
N0–N1	38	33	5	2.796	0.094	25	13	2.179	0.139
N2–N3	58	42	16			46	12		
TNM stage									
I–II	27	25	2	1.207	0.271	18	9	1.812	0.178
III–IV	69	58	11			55	14		
Tumor histology									
SQC	44	39	5	0.676	0.411	30	14	2.013	0.155
ADC	52	43	9			42	10		
Tumor differentiation									
Well/moderate	69	60	9	0.066	0.796	52	17	0.251	0.616
Poor	27	24	3			19	8		

Note: *p* values were derived from the χ^2^ test.

## Data Availability

The datasets and data used in this study can be obtained from official website or corresponding author.

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
