# Peer review of "Stearoyl-CoA Desaturases1 Accelerates Non-Small Cell Lung Cancer Metastasis by Promoting Aromatase Expression to Improve Estrogen Synthesis"

_ijms, 2023, doi:10.3390/ijms24076826_

Round 1

Reviewer 1 Report

Recommendation: publish after major revision

The manuscript entitled "Stearoyl-CoA desaturases1 accelerates non-small cell lung cancer metastasis by promoting aromatase expression to improve estrogen synthesis" by Chen et al shows that targeting lipid metabolism through the inhibition of the key enzyme SCD1 is a promising anti-cancer therapy for NSCLC. SCD1 knockdown reduced tumor cell migration and invasion by regulating the β-catenin/CYP19A1 pathway and reducing estrogen synthesis. Overall, the study is well-designed and the results are significant. However, there are some points that need to be addressed before the manuscript can be accepted for publication.

Major

1. The authors mentioned that "XAV939 treatment blocked the increase in CYP19A1 expression induced by SCD1 overexpression." Can XAV939 directly inhibit CYP19A1 expression in PC9, HCC827 or other NSCLC cells?

2. Does CYP19A1 knockdown or inhibition affect the mRNA and protein expression of SCD1?

3. Is the induction of CYP19A1 expression by SCD1 due to an increase in estrogen levels? The study did not verify whether estrogen affects CYP19A1 expression and the Wnt/β-catenin signaling pathway.

4. Can the SCD1 inhibitor A939572 inhibit tumor growth in animal models?

Minor 

5. The captions of Figure 1A&B 1C are reversed.

6. The spelling of "overall survival" in Figure 1E should be corrected.

7. Line 292 on page 9 needs to be revised to correct the error "SCD1the expression."

8. The English language of the manuscript needs further editing and improvement as there are several grammatical issues.

Reviewer 2 Report

The experiments look well carried out and the manuscript is already correct. I have some observations that the authors can bear in mind for re-structuring the rationale and/or background. With those comments I don’t ask for additional experiments, only to reason based on their results.

  1. Nuclear factor kB(NF-kB) directly regulates expression levels of lipid desaturases (Li et al. Cell Stem Cell 2017).
  2. Only EGFR mutant cells have been used (2 cell lines). Notwithstanding, lung cancer cells with STK11and/or KEAP1 loss have over-expression of SCD1 causing resistance to ferroptosis (Wohlhieter et al. Cell Reports 2020). SCD1 Inhibition causes tumor growth inhibition in KRAS mutant lung cancer cell. Synergism was found with the combination of SCD1 plus ferroptosis inducer.
  3. Could you find a relationship between ER and SCD1 expression.
  4. Could you suggest any limitations in your study and likewise what are the clinical implications?

Round 2

Reviewer 1 Report

The authors have addressed all of my concerns and I recommend accepting this manuscript for publication.